# Osteopathy Referrals to and from General Practitioners: Secondary Analysis of Practitioner Characteristics from an Australian Practice-Based Research Network

**DOI:** 10.3390/healthcare12010048

**Published:** 2023-12-25

**Authors:** Brett Vaughan, Michael Fleischmann, Sandra Grace, Roger Engel, Kylie Fitzgerald, Amie Steel, Wenbo Peng, Jon Adams

**Affiliations:** 1Department of Medical Education, The University of Melbourne, Melbourne, VIC 3010, Australia; kylie.fitzgerald@unimelb.edu.au; 2School of Public Health, University of Technology Sydney, Sydney, NSW 2007, Australia; michael.fleischmann2@rmit.edu.au (M.F.); amie.steel@uts.edu.au (A.S.); wenbo.peng@uts.edu.au (W.P.); jon.adams@uts.edu.au (J.A.); 3Faculty of Health, Southern Cross University, Lismore, NSW 2480, Australia; sandra.grace@scu.edu.au (S.G.); roger.engel@mq.edu.au (R.E.); 4School of Health and Biomedical Science, RMIT University, Melbourne, VIC 3001, Australia; 5Department of Chiropractic, Macquarie University, Sydney, NSW 2000, Australia

**Keywords:** allied health occupations, general practice, health workforce, musculoskeletal pain, osteopathic medicine, primary health care, referral, consultation

## Abstract

Australian osteopaths engage in multidisciplinary care and referrals with other health professionals, including general practitioners (GPs), for musculoskeletal care. This secondary analysis compared characteristics of Australian osteopaths who refer to, and receive referrals from, GPs with osteopaths who do not refer. The analysis was undertaken to identify pertinent characteristics that could contribute to greater engagement between Australian osteopaths and GPs. Data were from the Australian osteopathy practice-based research network comprising responses from 992 osteopaths (48.1% response rate). Osteopaths completed a practice-based survey exploring their demographic, practice, and clinical management characteristics. Backward logistic regression identified significant characteristics associated with referrals. Osteopaths who reported sending referrals (*n* = 878, 88.5%) to GPs were more likely than their non-referring colleagues to receive referrals from GPs (aOR = 4.80, 95% CI [2.62–8.82]), send referrals to a podiatrist (aOR = 3.09, 95% CI [1.80–5.28]) and/or treat patients experiencing degenerative spinal complaints (aOR = 1.71, 95% CI [1.01–2.91]). Osteopaths reporting receiving referrals (*n* = 886, 89.3%) from GPs were more likely than their non-referring colleagues to send referrals to GPs (aOR = 4.62, 95% CI [2.48–8.63]) and use the Medicare EasyClaim system (aOR = 4.66, 95% CI [2.34–9.27]). Most Australian osteopaths who report engaging in referrals with GPs for patient care also refer to other health professionals. Referrals from GPs are likely through the Chronic Disease Management scheme. The clinical conditions resulting in referrals are unknown. Further research could explore the GP–osteopath referral network to strengthen collaborative musculoskeletal care. The outcomes of this study have the potential to inform Australian osteopaths participating in advocacy, public policy and engagement with Australian GPs.

## 1. Introduction

Increasing rates of chronic disease in Australia have resulted in calls for a more coordinated approach to patient care [1]. Musculoskeletal complaints affect nearly one in three Australians [2] and represent approximately 18–25% of patient visits to Australian general practitioners (GPs) [3,4]. Therefore, GPs are ideally placed to lead the coordination and management approaches in musculoskeletal care. The management of musculoskeletal complaints occurs largely in primary care [4,5,6] with a range of allied health professions, including osteopaths, assisting in care delivery [7,8].

Australian osteopaths treat an array of musculoskeletal conditions [9,10] with interventions such as manual therapy, exercise and education [9,11], all with a minimal adverse event profile [12,13]. Burke, Myers and Zhang [9] identified that 5–7% of patients who seek osteopathy care are referred by a GP [9,10]. These findings are also supported in more recent work reporting 40% of New South Wales-based GPs having referred patients to a chiropractor or osteopath at some point during the previous year [14]. Other reports of high utilisation rates for osteopathy in rural and regional Australia have also been previously reported [15].

Similar rates of patient referral to GPs by osteopaths have also been reported [9,10,16,17]. Australian osteopaths are included in third-party payment systems related to work and transport accident injuries, war veterans, and chronic disease management (CDM) plans [18] with some systems (war veterans, CDM plans) requiring the GP to initiate referral and facilitate the communication and/or coordination of patient care between GP and osteopath. Approximately 38% of patients presenting to Australian osteopaths experience another chronic disease [9], and referrals to GPs may also be associated with the management of these complaints.

The study presented here reports the first analysis of osteopathy referrals to and from general practice in the Australian healthcare system, utilising a nationally representative sample of osteopaths drawn from the Australian Osteopathy Research and Innovation Network (ORION) project. ORION is the largest voluntary national practice-based research network (PBRN) of osteopathy worldwide. The work updates previous Australian literature [9,10,19] drawing on larger samples of the profession and the exploration of other aspects of practice not captured in previous studies. In this study, we identify predictors for Australian osteopaths sending referrals to, or receiving referrals from, GPs. The results of this study will be useful for developing a better understanding of the referral relationship between Australian osteopaths and GPs, to potentially inform professional development and advocacy, and to inform public and primary healthcare policy.

## 2. Materials and Methods

The study is a secondary analysis of the ORION PBRN project baseline practitioner database [20]. The ORION PBRN was established and administered through the School of Public Health, University of Technology Sydney (Australia).

### 2.1. Participants

Participant recruitment for ORION was undertaken from July through to December 2016, and the questionnaire was completed during this time. Registered Australian osteopaths who were members of an osteopathy professional association were invited to participate. At the time of data collection, there were 2020 osteopaths registered in Australia and approximately 85% of the profession were members of the professional association, with the survey being sent to 1717 registered osteopaths. ORION recruited 992 osteopaths during the data collection period. The ORION database captured 49.1% of the total population of registered osteopaths at the time of recruitment and is nationally representative of the profession on a number of key parameters [20].

### 2.2. Data Collection

ORION project participants completed a 27-item questionnaire examining demographic, practice, and patient management characteristics. Demographic variables included practitioner age, gender, and practice location. Practice characteristics included average patient care hours and patient visits per week, number of practice locations, types of other health professionals active in the same practice location, referral relationships, practice location (urban/rural/remote), use of diagnostic imaging, clinical diagnosis and use of electronic records and beliefs about prescribing, in addition to patient management. Patient management characteristics included frequency of patient presentations, discussion of lifestyle behaviours, frequency of treating specific patient groups, and frequency of use of specific osteopathy techniques. The ORION questionnaire did not capture the reasons for referrals to and from GPs.

### 2.3. Data Analysis

The focus of this secondary analysis was on the following outcome variables/questions: ‘sending referrals to general practitioners’ (yes/no) and ‘receiving referrals from general practitioners’ (yes/no).

The exposure variables were the demographic and practice characteristics, and patient management was outlined in the *Data collection* description above. The response options to all variables about osteopaths’ clinical management and the variation in the frequency of referring people for diagnostic imaging were ‘never’, ‘rarely’, ‘sometimes’ and ‘often’. These response options were dichotomised to ‘not often’ (comprising ‘never, ‘rarely’ and ‘sometimes’) and ‘often’. The demographic variables were age, the number of years in private osteopathy practice, average patient care hours per week, and average patient visits per week and were included in the analyses as continuous variables. All other variables included in our analysis are reported in binary form (yes/no).

### 2.4. Statistical Analyses

Bivariate analyses were used to identify the associations between the outcome and exposure variables via chi-square and Fisher exact tests for categorical variables and *t*-tests for continuous variables. Multivariate logistic regression was used to identify predictors for sending to or receiving referrals from GPs. Variables with *p* < 0.20 in these bivariate analyses were included in a backward elimination, multivariate logistic regression analysis [21]. Adjusted odds ratios (aORs) with a 95% confidence interval (CI) and *p*-values were calculated from this regression modelling. The statistical significance was set at α = 0.05 and effect sizes (Cohen’s *d*) were calculated as required. Descriptive statistics and logistic regression were performed using SPSS (version 25).

### 2.5. Ethics Approval

Ethics approval for this study was granted by the University of Technology Sydney Human Research Ethics Committee (approval number: 2014000759). All participants provided written informed consent.

## 3. Results

Of the 992 responses to the ORION questionnaire, 878 (88.5%) osteopaths reported sending referrals to a GP and 886 (89.3%) reported receiving referrals from a GP. Respondents’ age, gender, years in clinical practice, patient care hours per week, patient consultations per week and involvement in non-clinical roles are reported according to sending (Table 1) or receiving GP referrals (Table 2). Osteopaths with a Master’s degree were more likely to send referrals to a GP (χ^2^ (5) = 12.03, *p* = 0.03) and receive referrals from a GP (χ^2^ (5) = 17.28, *p* < 0.01) compared to those who did not receive or send GP referrals.

### 3.1. Sending Referrals to Australian GPs

Osteopaths who report initiating referrals to GPs were five times more likely to receive referrals from GPs (unadjusted OR = 5.95) compared to osteopaths who did not initiate referrals to GPs (Appendix A). Osteopaths who referred patients to GPs were also four times more likely to refer to other health professionals including specialist medical practitioners (e.g., rheumatologist, paediatrician) (unadjusted OR = 4.10) and podiatrists (unadjusted OR = 4.62).

Regarding patient assessment, osteopaths who referred patients to GPs were nearly five times more likely to use orthopaedic testing (unadjusted OR = 4.93) and more than twice as likely to use neurological testing (unadjusted OR = 2.10) and/or cranial nerve testing (unadjusted OR = 2.43) compared to osteopaths who did not refer patients to GPs (Appendix A). Osteopaths who initiated referral to GPs were twice as likely to treat degenerative spinal conditions (unadjusted OR = 2.00) and/or patients aged 4–18 years (unadjusted OR = 2.12) (Appendix A) compared to osteopaths who did not refer patients to GPs.

### 3.2. Receiving Referrals from Australian GPs

Australian osteopaths who reported receiving referrals from GPs were over 1.5 times more likely to refer to and receive referrals from other allied health professionals compared to osteopaths who did not refer patients to GPs (Appendix A). Osteopaths who reported receiving referrals from GPs more likely to use diagnostic imaging for investigating fractures (unadjusted OR = 1.45) and unknown pathologies (unadjusted OR = 1.47) compared to those osteopaths who reported not receiving referrals from GPs. Australian osteopaths who reported receiving referrals from GPs were more than twice as likely to report treating patients over 65 years of age (unadjusted OR = 2.44), compensable work injury patients (unadjusted OR = 4.36) and/or non-compensable traffic injury patients (unadjusted OR = 2.75) compared to those osteopaths not receiving GP referrals (Appendix A).

Osteopaths initiating patient referral to a GP were significantly more likely to report sending referrals to four health professionals (aOR range: 2.04–3.93) and/or the use of cranial nerve assessment in patient care (aOR = 2.00) (Table 3). Receiving GP referrals was significantly associated with osteopaths sending referrals to GPs (aOR = 4.62) and the use of Medicare EasyClaim (electronic rebate claim system) (aOR = 4.66) (Table 3).

## 4. Discussion

Our analyses, drawn from a large, nationally representative sample of Australian osteopaths, suggests a substantial volume of patient traffic between general practitioners and osteopaths. A key finding of our analysis is that Australian osteopaths who refer patients to GPs are more than twice as likely to also refer patients to other allied health professionals and medical specialists compared to those who do not send referrals to GPs. This finding is consistent with results from a regional practice investigation which previously identified osteopaths as regularly reporting referrals to other health professionals [9,19]. The results of the current secondary analysis suggest that Australian osteopaths are engaging in multidisciplinary care via a referral network that includes GPs [22]. Osteopaths who refer to GPs may be more aware of their scope of practice (i.e., musculoskeletal focus) and of clinical practice guidelines for such common musculoskeletal conditions as low back pain [23,24,25] that recommend referral to other health professionals as appropriate to optimise patient outcomes.

Those osteopaths in our study who reported referring patients to GPs were nearly six times as likely to receive patient referrals from GPs. This finding suggests a possible reciprocity in the GP–osteopath referral relationship [26]. Work by Wardle, Sibbritt and Adams [14] in rural and regional New South Wales reported GPs were 67% more likely to refer to an osteopath or chiropractor if they had observed positive patient outcomes. Similarly, it may be that the referral reciprocity identified in our work is related to GPs’ observing positive patient outcomes from osteopathy care and/or that patients have requested the referral [14,26,27,28] or both services are readily available to patients [14,29]. Further work exploring the nature of these referrals is therefore warranted.

Our work highlights some of the practice characteristics that may underpin the referral relationship between GPs and osteopaths. Australian osteopaths who reported patient referral to GPs were more likely to use orthopaedic and neurological assessments as part of their diagnostic approach compared to osteopaths who did not report referring. Such a finding may be of particular significance in that communicating patient assessments in a format and language that is familiar to GPs may help further the GP’s understanding of the nature of the referral. The use of a common language may also facilitate patient referrals [26,30]. Referrals from osteopaths may also lead to developing an appreciation of the potential role of osteopaths in primary healthcare [31] and build trust between the two health professions [26]. Further work is needed to investigate the details of how Australian GPs and osteopaths communicate with regard to referrals and the diagnoses associated with these referrals, potentially using qualitative approaches to explore communication strategies and clinical reasoning.

Those osteopaths in our study who reported receiving patient referrals from GPs were six times more likely to ‘often’ use the electronic Medicare system Medicare EasyClaim than those osteopaths who did not report receiving GP referrals. The Medicare EasyClaim system allows patients to claim a rebate for their osteopathy treatment under the Australian public healthcare system. Patients referred by their GP via a Chronic Disease Management (CDM) plan are entitled to a rebate on up to five consultations with an osteopath per year [32]. Our results suggest some referrals from GPs to Australian osteopaths may be for CDM plans where Medicare EasyClaim is used to facilitate the patient’s rebate for treatment. Orrock [10] and Burke, Myers and Zhang [9] reported that 4–8% of the total number of patients treated by Australian osteopaths were referred from GPs. We can infer from our results that this figure is likely to be an underestimation, as there has been a significant growth in the total number of consultations under the CDM scheme in the past 5 years [33]. Further research should explore the number of referrals under the CDM scheme using publicly available data in addition to using both qualitative and quantitative approaches to explore the types of conditions that GPs are referring patients to osteopaths with and the outcomes of osteopathy care for these patients.

Australian osteopaths who reported practising in a rural or remote location were 50% more likely to receive referrals from a GP than those osteopaths practising in an urban location. This finding is consistent with earlier work by Wardle, Sibbritt and Adams [14] who reported a significant referral relationship between osteopaths and GPs in rural areas of New South Wales (Australia). Such outcomes may be due to there being a smaller network of health professionals accessible in many rural areas [14,29], increased GP knowledge of other health professionals in a town or rural locale [30], and possibly patient preference for care by a particular health professional [14]. The data in the current study do not allow for commentary on whether these factors account for the increased probability of referrals, but our findings do suggest the need for further work to explore why referrals from GPs to osteopaths in rural and regional settings differ for urban environments using surveys or practice audit data.

There are several limitations associated with our study. The self-reported data are subject to recall and social desirability bias [34], and we are unable to ascertain how osteopaths interpreted the items on the original ORION questionnaire, and this may have affected our results. We have made several inferences based on the strength of the odds ratios that are solely related to the individual osteopath’s perception of their practice. The extent to which these responses are consistent with the perceptions of GPs [35] requires additional investigation. The age of the data may also be a limitation of the current work. It is anticipated that a new ORION practice survey will be distributed in 2024, and this may provide different outcomes given the COVID pandemic and other health system changes that have occurred since data collection in 2016.

Notwithstanding these limitations, this study provides insights into the bi-directional referral relationship between osteopaths and GPs in Australia via a large, nationally representative sample of osteopaths. Future research should be directed toward developing a greater understanding of the conditions resulting in, and motivations for, GP referrals to osteopaths, particularly through the CDM scheme. This work has the potential to guide coordinated care in the primary care context. Work should also explore the effectiveness of osteopathy care under this scheme, including patient outcomes and cost-effectiveness.

## 5. Conclusions

The majority of Australian osteopaths report sending referrals to and receiving referrals from GPs in addition to engaging in referrals with a range of other health professionals. The current study suggests that the practice characteristics of Australian osteopaths who receive and send referrals to GPs, including the use of examination approaches, may foster a shared understanding of the role of osteopathy care in managing musculoskeletal complaints. There is also the possibility of osteopaths identifying other chronic presentations where referral to a GP is warranted potentially for co-management or additional investigation. Further research is required to help understand the process and patient cohort involved in bi-directional osteopath–general practitioner referrals and the nature and effectiveness of such multidisciplinary patient care.

## Figures and Tables

**Table 1 healthcare-12-00048-t001:** Practitioner characteristics of Australian osteopaths who report sending referrals to general practitioners.

	Yes	No	*p*-Value
	(*n* = 878, 88.5%)	(*n* = 114, 11.5%)	
Gender			
Male	514 (58.5%)	62 (54.4%)	0.40
Female	364 (41.5%)	52 (45.6%)	
Highest educational qualification			
Diploma	48 (4.8%)	14 (1.4%)	0.03
Advanced diploma	7 (0.2%)	2 (0.2%)	
Bachelor’s degree	201 (20.3%)	17 (1.7%)	
Master’s degree	601 (60.6%)	80 (8.1%)	
PhD	5 (0.5%)	0	
Involved in as an osteopath			
University teaching	109 (11.0%)	7 (0.7%)	0.05
Clinical supervision	138 (13.9%)	12 (1.2%)	0.14
Professional organisations	96 (9.7%)	11 (1.1%)	0.68
Research	52 (5.2%)	2 (0.2%)	0.08
Volunteer	148 (14.9%)	11 (1.1%)	0.05
Age (years)			
Mean (± SD)	37.6 (± 10.3)	40.8 (± 14.33)	0.13
Years in clinical practice			
Mean (± SD)	11.2 (± 8.5)	13.2 (± 12.0)	0.49
Patient care hours per week			
Mean (± SD)	27.8 (± 12.0)	28.6 (± 12.7)	0.57
Patient visits per week			
Mean (± SD)	36.1 (± 17.9)	38.8 (± 23.0)	0.19

**Table 2 healthcare-12-00048-t002:** Practitioner characteristics of Australian osteopaths who report receiving referrals from general practitioners.

	Yes	No	*p*-Value
	(*n* = 886, 89.3%)	(*n* = 106, 10.7%)	
Gender			
Male	510 (57.6%)	66 (62.3%)	0.35
Female	376 (42.4%)	40 (37.7%)	
Highest educational qualifications			
Diploma	50 (5.0%)	12 (1.2%)	<0.01
Advanced diploma	6 (0.6%)	3 (0.3%)	
Bachelor’s degree	206 (20.8%)	12 (1.2%)	
Master’s degree	603 (60.8%)	78 (7.9%)	
PhD	5 (0.5%)	0	
Involved in as an osteopath			
University teaching	106 (10.7%)	10 (1.0%)	0.44
Clinical supervision	137 (13.8%)	13 (1.3%)	0.38
Professional organisations	98 (9.9%)	9 (0.9%)	0.42
Research	51 (5.1%)	3 (0.3%)	1.00
Volunteer	147 (14.8%)	12 (1.2%)	0.16
Age (years)			
Mean (± SD)	37.8 (± 10.3)	39.6 (± 13.7)	0.72
Years in clinical practice			
Mean (± SD)	11.3 (± 8.7)	12.0 (± 11.3)	0.58
Patient care hours per week			
Mean (± SD)	28.1 (± 11.3)	26.5 (± 13.1)	0.18
Patient visits per week			
Mean (± SD)	36.8 (± 18.1)	33.8 (± 22.2)	0.15

**Table 3 healthcare-12-00048-t003:** Adjusted odds ratios (aOR) for characteristics of Australian osteopaths who report sending referrals to and receiving referrals from general practitioners.

Sending Referrals to General Practitioners	aOR	95% CI	*p*-Value
Patient visits per week	0.98	0.97, 0.99	0.028
Co-located with other osteopaths	2.00	1.17, 3.45	0.011
Co-located with occupational therapist	0.20	0.05, 0.79	0.022
Co-located with an acupuncturist	0.38	0.21, 0.70	0.002
Send referrals to medical specialists	3.93	2.07, 7.47	<0.01
Send referrals to a podiatrist	3.09	1.80, 5.28	<0.01
Send referrals to a physiotherapist	2.04	1.04, 4.00	0.038
Send referrals to a naturopath	2.42	1.35, 4.34	0.003
Receive referrals from a general practitioner	4.80	2.62, 8.82	<0.01
Use cranial nerve testing to assist with diagnosis	2.00	1.17, 3.43	0.012
Often treat degenerative spinal complaints	1.71	1.01, 2.91	0.046
Often treat headaches	0.18	0.05, 0.62	0.006
Often treat patients aged 4–18 years	2.03	1.06, 3.89	0.034
**Receiving referrals from general practitioners**			
Send referrals to a general practitioner	4.62	2.48, 8.63	<0.01
Send referrals to a podiatrist	0.52	0.29, 0.95	0.034
Receive referrals from a medical specialist	3.59	1.38, 9.34	0.009
Receive referrals from a podiatrist	2.68	1.41, 5.10	0.002
Receive referrals from a massage therapist	2.03	1.18, 3.46	0.010
Often use Medicare EasyClaim	4.66	2.34, 9.27	<0.01
Often treat patients aged 65 years or older	1.78	1.05, 2.99	0.030

## Data Availability

Data underpinning the study are available on reasonable request from the Australian Research Consortium in Complementary and Integrative Medicine (arccim@uts.edu.au).

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
