# Peer review of "Osteopathy Referrals to and from General Practitioners: Secondary Analysis of Practitioner Characteristics from an Australian Practice-Based Research Network"

_healthcare, 2023, doi:10.3390/healthcare12010048_

Round 1
Reviewer 1 Report
Comments and Suggestions for Authors
This very well-written article reports the characteristics of osteopaths who are more likely to engage in a referring relationship with GPs.
I will provide feedback in a structured way to help the authors to strengthen the already well-conducted report.
Generalities:
The only real possible big issue related to the presented data is that they are almost 8 years old and using them as a possible insight into today's situation might be misleading. I will limit the review to highlighting this issue leaving any decision on the suitability for publication to the editor since I will only focus on areas of improvement from a methodological and reporting perspective.
Abstract:
• I suggest reporting the attrition rate with indications of how (eventually) incomplete questionnaires were handled
• It is not clear to me why referrals between osteopaths and GP and vice versa should only strengthen musculoskeletal care and not secondary prevention in its entirety as well. I suggest broadening the proposed field including the possible impact on fastening the diagnosis of underlying pathologies (not necessarily limited to the msk system).
Introduction
• The study rationale might be better highlighted. I suggest answering the question "Why is it important to analyze these data? what can we understand better after the analysis? what impact can it have on the profession?"
Methods:
• I suggest reporting the attrition rate with indications of how (eventually) incomplete questionnaires were handled
• I suggest reporting if there was a system to monitor double respondents and how
Results:
• Line 205-207 it is unclear to me why using a common language can only impact referrals related to musculoskeletal disorders
Conclusion:
• Line 257: This is a bit of an overstatement. From what I can see it is not the “majority of Australian osteopaths”, it is the majority of respondents (which are anyway less than half of the sample, which I am not sure anyway represents the entire target population).
• Line 261-262 It is unclear to me why there are no references to secondary prevention and why there is a specification for musculoskeletal complaints.
Author Response
Abstract:
I suggest reporting the attrition rate with indications of how (eventually) incomplete questionnaires were handled
Thank you for the comment. The study is a secondary analysis of data collected as part of the Australian osteopathy practice-based research network. The design and data collection have been described in both of the following papers:
Adams J, Sibbritt D, Steel A, Peng W. A workforce survey of Australian osteopathy: analysis of a nationally-representative sample of osteopaths from the Osteopathy Research and Innovation Network (ORION) project. BMC Health Services Research. 2018 Dec;18:1-7.
Steel A, Peng W, Sibbritt D, Adams J. Introducing national osteopathy practice-based research networks in Australia and New Zealand: an overview to inform future osteopathic research. Scientific Reports. 2020 Jan 21;10(1):846.
The manuscript is already at the word limit so we are reluctant to add more detail. These details can be found in the papers referenced above. That said, we are happy to be guided by the Editors if these details should be included in the manuscript.
It is not clear to me why referrals between osteopaths and GP and vice versa should only strengthen musculoskeletal care and not secondary prevention in its entirety as well. I suggest broadening the proposed field including the possible impact on fastening the diagnosis of underlying pathologies (not necessarily limited to the msk system).
Thank you for the comment and suggestion. This is a helpful suggestion and we have expanded on other chronic diseases that patients may present with to osteopaths in Australia at lines 58-60. The scope of practice of osteopathy in Australia is predominantly musculoskeletal diagnosis care so we have been careful to not go beyond this in our descriptions.
Introduction
The study rationale might be better highlighted. I suggest answering the question "Why is it important to analyze these data? what can we understand better after the analysis? what impact can it have on the profession?"
Thank you for this suggestion. We have added at lines 68-71 “The results of this study will be useful for developing a better understanding of the referral relationship between Australian osteopaths and GPs, potentially inform professional development and advocacy, and inform public and primary healthcare policy.”
Methods:
I suggest reporting the attrition rate with indications of how (eventually) incomplete questionnaires were handled
I suggest reporting if there was a system to monitor double respondents and how
As per the response to the first comment, we have directed readers to the following papers where the data collection strategy is detailed:
The design and data collection have been described in both of the following papers:
Adams J, Sibbritt D, Steel A, Peng W. A workforce survey of Australian osteopathy: analysis of a nationally-representative sample of osteopaths from the Osteopathy Research and Innovation Network (ORION) project. BMC Health Services Research. 2018 Dec;18:1-7.
Steel A, Peng W, Sibbritt D, Adams J. Introducing national osteopathy practice-based research networks in Australia and New Zealand: an overview to inform future osteopathic research. Scientific Reports. 2020 Jan 21;10(1):846.
As mentioned above, we are happy to be guided by the Editors if these details should be included in the manuscript.
Results:
Line 205-207 it is unclear to me why using a common language can only impact referrals related to musculoskeletal disorders
Thank you for the comment. We have amended lines 226-227 to read “The use of a common language may also facilitate patient referrals. Referrals from osteopaths may also lead to developing an appreciation of the potential role of osteopaths in primary healthcare and build trust between the two health professions.”
Conclusion:
Line 257: This is a bit of an overstatement. From what I can see it is not the “majority of Australian osteopaths”, it is the majority of respondents (which are anyway less than half of the sample, which I am not sure anyway represents the entire target population).
Thank you for the comment. The sample in the current work was nationally-representative of the sample at the time of data collection as reported in the following cross-sectional work:
Adams J, Sibbritt D, Steel A, Peng W. A workforce survey of Australian osteopathy: analysis of a nationally-representative sample of osteopaths from the Osteopathy Research and Innovation Network (ORION) project. BMC Health Services Research. 2018 Dec;18:1-7.
The majority of respondents reported sending referrals to (88.5%) and receiving referrals from (89.3%) general practitioners. The nationally representative nature of the sample supports our interpretation with respect to the “majority of Australian osteopaths”. As such, we have retained this aspect of the manuscript.
Line 261-262 It is unclear to me why there are no references to secondary prevention and why there is a specification for musculoskeletal complaints.
Thank you for the comment. We have focused on musculoskeletal complaints as this is the dominant scope of practice for Australian osteopaths based on their training. With respect to other complaints we have added at lines 58-60 “Approximately 38% of patients presenting to Australian osteopaths experience another chronic disease [5] and referrals to GPs may also be associated with the management of complaints” and at lines 296-298 “There is also the possibility of osteopaths identifying other chronic presentations where referral to a GP is warranted, potentially for co-management or additional investigation.”
Reviewer 2 Report
Comments and Suggestions for Authors
Thank you for this interesting paper which highlights an important issue in practice.
I have added some minor queries and comments below which I hope are helpful.
Methods
Can you add when the data were collected. You mention that recruitment occurred July to Dec, 2016 but when was the actual survey distributed.
Results
Sending referrals to Australian GPs
Osteopaths who referred patients to GPs were also four times more likely to refer to other health professionals including specialist medical practitioners (unadjusted OR=4.10) and podiatrists (unadjusted OR=4.62). Can you give some examples of specialist medical professionals for clarity.
4. Discussion
The results of the current secondary suggests that – missing word here
This previous work suggested such outcomes may be 231
due to their being a smaller network of health professionals accessible in many rural areas, 232 – spelling change needed
Additional areas for consideration
You’ve identified various areas for future research but perhaps you could elaborate on some of the methodological approaches you could employ for these.
In the limitations you don’t discuss the fact that the data comes from members who have joined ORION and may feel more confident about their practice and the effect this could have on the data.
The data appears to be quite old now. Can you comment on plans to update the findings especially in view of the effects of the pandemic and how this has affected osteopathic practice, GP referrals, and patients being able to access care generally in healthcare providers in Australia.
Author Response
Methods
Can you add when the data were collected. You mention that recruitment occurred July to Dec, 2016 but when was the actual survey distributed.
Thank you for the comment. We have clarified this at line 79.
Results
Sending referrals to Australian GPs
Osteopaths who referred patients to GPs were also four times more likely to refer to other health professionals including specialist medical practitioners (unadjusted OR=4.10) and podiatrists (unadjusted OR=4.62). Can you give some examples of specialist medical professionals for clarity.
Thank you for the suggestion. We have added two examples, rheumatologist and paediatrician, at line 163.
- Discussion
The results of the current secondary suggests that – missing word here
Amended to include ‘analysis’
This previous work suggested such outcomes may be 231
Amended to ‘These authors’
due to their being a smaller network of health professionals accessible in many rural areas, 232 – spelling change needed
Amended to ‘there’
Additional areas for consideration
You’ve identified various areas for future research but perhaps you could elaborate on some of the methodological approaches you could employ for these.
Thank you for the comment. We have added suggested methodological approaches at lines 231-232, 247-248 and 270.
In the limitations you don’t discuss the fact that the data comes from members who have joined ORION and may feel more confident about their practice and the effect this could have on the data.
Thank you for the suggestion. We have included social desirability bias as one of the limitations at line 273.
The data appears to be quite old now. Can you comment on plans to update the findings especially in view of the effects of the pandemic and how this has affected osteopathic practice, GP referrals, and patients being able to access care generally in healthcare providers in Australia.
Thank you for the suggestion. We have added at lines 278-280 “It is anticipated that a new ORION practice survey will be distributed in 2024 and this may provide different outcomes given the COVID pandemic and other health system changes that have occurred since data collection in 2016.”
Reviewer 3 Report
Comments and Suggestions for Authors
Abstract:
- It is not clear what the authors are trying to explore and what the objective of the study is.
- What is the significance or clinical implications of this study?
- How the understanding of the referral pattern to and from the osteopath can shape or improve the clinical practice is not clear.
Introduction:
- What is the study hypothesis and research question clearly articulated?
- What is the state of the existing literature and what were the weaknesses in those studies that this study tries to answer?
Methods:
Data collection
- Did the authors collect information about the main reason/ complaint/ symptoms for the referrals to and from the osteopathic practitioners?
Statistical analyses
- “Multivariate logistic regression was used to identify predictors for sending to or receiving referrals from GPs”.
- This statement reflects the study objective and should be clearly stated in the introduction and the method section.
The authors did not explain how the goodness of fit was assessed and outliers in the model were assessed.
Results and discussions:
- Without a clear idea about the research question and hypothesis, it is difficult to interpret the results and their clinical relevance.
Author Response
Abstract:
- It is not clear what the authors are trying to explore and what the objective of the study is.
Thank you for the comment. We have now added the objective to the abstract at line 17.
- What is the significance or clinical implications of this study?
Thank you for the comment. We have now added the objective to the abstract at lines 32-33.
- How the understanding of the referral pattern to and from the osteopath can shape or improve the clinical practice is not clear.
Thank you for the comment. We have now added the objective to the abstract at lines 32-33.
Introduction:
- What is the study hypothesis and research question clearly articulated?
Thank you for the comment. The current work is exploratory and we have not proposed a hypothesis. The research question is described at lines 67-71.
- What is the state of the existing literature and what were the weaknesses in those studies that this study tries to answer?
Thank you for the query. We have included the following at lines 65-67 “The work updates previous Australian literature [5,6,15] drawing on larger samples of the profession and exploration of other aspects of practice not captured in these studies.”
Methods:
Data collection
- Did the authors collect information about the main reason/ complaint/ symptoms for the referrals to and from the osteopathic practitioners?
Thank you for the query. We noted in the abstract that complaint etc. was not collected and we have now reiterated this in the methods at lines 104-105.
Statistical analyses
- “Multivariate logistic regression was used to identify predictors for sending to or receiving referrals from GPs”.
- This statement reflects the study objective and should be clearly stated in the introduction and the method section.
Thank you for the comment. As identified above, the objective of the study has now been clarified.
The authors did not explain how the goodness of fit was assessed and outliers in the model were assessed.
Thank you for the comment. The purpose of the study was to identify potential predictors of referral rather than develop a prediction model. As we did not set out to develop a model, we did not evaluate goodness of fit and outliers.
Results and discussions:
- Without a clear idea about the research question and hypothesis, it is difficult to interpret the results and their clinical relevance.
Thank you for the comment. We hope that the amendments have addressed this concern.